
# A fast SWIR imager for observations of transient features in OH airglow

P. Hannawald[1], C. Schmidt[2], S. Wüst[2], and M. Bittner[1,2]

[1]Institute of Physics, University of Augsburg, Augsburg, Germany
[2]German Remote Sensing Data Center, German Aerospace Center, Oberpfaffenhofen, Germany

Received: 7 December 2015 – Accepted: 11 December 2015 – Published: 14 January 2016

Correspondence to: P. Hannawald (patrick.hannawald@physik.uni-augsburg.de)

Published by Copernicus Publications on behalf of the European Geosciences Union.

Discussion Paper | Discussion Paper | Discussion Paper | Discussion Paper

**AMTD**

doi:10.5194/amt-2015-382

A fast SWIR imager for observations of transient features in OH airglow

P. Hannawald et al.

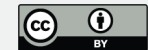

## Abstract

Since December 2013 the new imaging system FAIM (Fast Airglow IMager) for the study of smaller-scale features (both in space and time) is in routine operation at the NDMC (Network for the Detection of Mesospheric Change) station at DLR (German Aerospace Center) in Oberpfaffenhofen (48.1° N, 11.3° E).

Covering the brightest OH vibrational bands between 1 and 1.7 μm, this imaging system can acquire two frames per second. The field of view is approximately 55 km times 60 km at the mesopause heights. A mean spatial resolution of 200 m at a zenith angle of 45° and up to 120 m for zenith conditions are achieved. The observations show a large variety of atmospheric waves.

This paper introduces the instrument and compares the FAIM data with spectrally resolved GRIPS (GRound-based Infrared P-branch Spectrometer) data. In addition, a case study of a breaking gravity wave event, which we assume to be associated with Kelvin-Helmholtz-instabilities, is discussed.

## 1 Introduction

The OH airglow layer is located at a height of about 87 km with a half-width of approximately 4 km (e.g. Baker and Stair, 1988). It results from different chemical reactions leading to the emission of many vibrational-rotational lines in the visual and infrared optical range (for further details see e.g. Meinel, 1950; Roach and Gordon, 1973; Rousselot et al., 2000). Observing the infrared emissions of the vibrational-rotational excited OH molecules offers an unique possibility for studying atmospheric dynamics. Especially atmospheric gravity waves are prominent features in the measurements.

Due to density decreasing exponentially with altitude, the wave amplitude of upward propagating waves increases as long as the wave is not dissipating energy. Therefore, small wave amplitudes in the tropopause can reach high amplitudes in the mesosphere and lower thermosphere region (MLT). It is widely accepted that propagating atmo-

Discussion Paper | Discussion Paper | Discussion Paper | Discussion Paper | Discussion Paper |

# AMTD

doi:10.5194/amt-2015-382

**A fast SWIR imager for observations of transient features in OH airglow**

P. Hannawald et al.

spheric gravity waves are important for the understanding of atmospheric dynamics and the energy budget of the atmosphere as they provide the majority of the momentum forcing that drives the circulation in the MLT (Smith, 2012). A general overview about gravity waves and their importance for middle atmospheric dynamics can be found beside others in Fritts and Alexander (2003).

In order to acquire knowledge about gravity waves in this region, only a limited number of instruments can be used. Radars and lidars for example measure changes in temperature and wind with a comparatively high vertical resolution (see e.g. Sato, 1994; Chu et al., 2005; Dunker et al., 2015).

Beside these active instruments, passive spectrometers and imagers are frequently used to observe airglow emissions originating in the MLT region and being modulated by gravity waves. The Network for the Detection of Mesospheric Change (NDMC) for example currently lists 57 observing sites, some of which are equipped with more than just one of these instruments. Spectrometers used for airglow observations are either space-born (e.g. ENVISAT-SCIAMACHY, Bovensmann et al., 1999; von Savigny, 2015, and references therein) or ground-based (e.g. Nakamura et al., 1999; Schmidt et al., 2013; Noll et al., 2014). Airglow spectra are usually used for determining the atmospheric temperature at the height of the emission layer.

The spectral range of imaging instruments covers a wide range of airglow emissions, mostly some bandpass between 500 and 1700 nm depending on the sensor material and applied filters. Many of these systems provide a large field of view (FOV) using fisheye lenses (see e.g. Taylor et al., 1995; Shiokawa et al., 1999; Smith et al., 2009; Mukherjee et al., 2010). Others are using a smaller aperture to focus on a distinct part of the nightsky (e.g. Hecht et al., 2002; Moreels et al., 2008). The temporal resolution typically varies from one image every few seconds up to few minutes (see e.g. Hecht et al., 2002; Taguchi et al., 2004). Another type of instrument related to both, spectrometers and imagers, is the MTM (Mesospheric Temperature Mapper, Taylor et al., 1999) and the AMTM (Advanced Mesospheric Temperature Mapper, Pautet et al., 2014), both

Discussion Paper | Discussion Paper | Discussion Paper | Discussion Paper |

**AMTD**

doi:10.5194/amt-2015-382

**A fast SWIR imager for observations of transient features in OH airglow**

P. Hannawald et al.

being imagers using very narrow band filters to isolate individual emission lines, allowing the determination of airglow temperature during later processing.

The imager presented here provides a high temporal and spatial resolution in order to focus on small scale and transient phenomena. In Sect. 2, the instrument and its setup is described and observations are compared to the infrared spectrometer system GRIPS 13, measuring parallel to the imager at the same site. The data analysis is explained in Sect. 3, while the results of a case study are presented and discussed in Sect. 4.

## 2 Instrumentation and operation

Unlike most of the existing airglow imagers, it is the primary goal to acquire images of the airglow layer at the fastest rate possible. Therefore, the system is designed to observe some of the brightest hydroxyl emission bands in the short wave infrared between approximately 1400–1650 nm utilizing a sensitive detector as well as optics with high transmission.

The 320 to 256 pixel sensor array of the FAIM 1-Instrument (Fast Airglow IMager 1) based on InGaAs has a spectral responsivity in the range from 950 to 1650 nm (model "Xeva" manufactured by Xenics nv). It is equipped with a three stage thermoelectric cooler and usually cooled to 235 K to reduce the dark current. Similar detectors are used by Pautet et al. (2014). The standard optics consist of an "F# 1.4 Schneider-Kreuznach SWIRON" lens with a focal length of 23 mm. In front of the lens is a mechanical shutter for protecting the sensor from aging processes due to direct sunlight at daytimes (see sketch in Fig. 1 for the instrument's setup).

In this setup standard exposure times of only 500 ms are realized. The images are stored continuously with a delay between two consecutive images due to readout and processing of about 10 ms. This high temporal resolution enables the study of transient features in the airglow, providing the possibility to observe phenomena with frequencies

**AMTD**

doi:10.5194/amt-2015-382

**A fast SWIR imager for observations of transient features in OH airglow**

P. Hannawald et al.

significantly higher than the Brunt-Väisälä-frequency, e.g. infrasound (see e.g. Pilger et al., 2013) or turbulence.

The signal of the airglow is converted into 12-bit greyscale images with values ranging from 0 to 4095, hereafter denoted as counts or arbitrary units (au). A series of 100 darkened images is acquired in order to determine the noise with the current settings. The standard deviation of the mean value of each image results in 25 counts.

Before analysis, all images are flat-fielded with images measured in front of a homogeneous large area black body source. This procedure eliminates the so-called fixed pattern noise as well as the vignetting of the lens.

The setup of FAIM 1 is shown in Fig. 2. The instrument is operated at a zenith angle of 45°. The lens provides a field of view (FOV) of 20° × 24° with a barrel distortion of less than 1 % which is neglected furthermore. Due to the fact that the instrument is not looking into zenith direction, the resulting area observed is trapezium-shaped with a size of about 55 km × 60 km in the altitude of the OH-emission peak in 87 km ± 4 km (height according to Baker and Stair, 1988). The instrument is located at Oberpfaffenhofen, Germany (48.09° N, 11.28° E) with an azimuth angle of 214° ± 1° (direction SSW). The observed area is located above parts of Southern Germany and the Austrian Alpine region (see Fig. 4c).

The observed trapezium shaped area of the airglow layer is mapped on the rectangular shaped sensor by reason of the measurement geometry. As a result of this, the images show a distorted view of the airglow layer and have to be corrected to get an equidistant scale, which is necessary for the following analyses. Therefore, a transformation is applied on the images to remap the pixels to the original shape. This transformation based on some trigonometrical considerations is mainly dependent on the zenith angle and the FOV. The left-hand side of Eq. (1) gives the geographical (Cartesian) coordinates depending on the zenith angle $\theta \in (45° \pm \frac{19.5°}{2})$ and the azimuth angle

**AMTD**

doi:10.5194/amt-2015-382

**A fast SWIR imager for observations of transient features in OH airglow**

P. Hannawald et al.

**AMTD**

doi:10.5194/amt-2015-382

**A fast SWIR imager for observations of transient features in OH airglow**

P. Hannawald et al.

$\phi \in (\pm\frac{24.1^\circ}{2})$ for each pixel:

$$\begin{pmatrix} x(y(z,\theta),z,\phi) \\ y(z,\theta) \\ z \end{pmatrix} = \begin{pmatrix} \sqrt{y^2+z^2}\cdot\tan(\phi) \\ z\cdot\tan(\theta) \\ 87\,\text{km} \end{pmatrix}, \tag{1}$$

with Cartesian coordinates $x$, $y$ and $z$; $y$ is parallel to the line of sight and $x$ is perpendicular to it (compare Fig. 2); the airglow layer is assumed to be of constant altitude $z = 87$ km. The origin is given by the location of the instrument.

After the geographical coordinates for each pixel have been determined, the area covered by the entire image is calculated. It amounts to approximately 3400 km$^2$, or 60 km × 55 km (height and width of the trapezium). For the 320 × 256 pixel array used, this refers to an approximate resolution of 200 m × 200 m (±10 %) per Pixel. The relation between the zenith angle and the mean spatial resolution is shown in Fig. 3 as thick solid line. The grey dashed line shows the size of the observed area. the standard zenith angle of 45° is marked by thin dashed lines. In zenith direction, the mean spatial resolution is 120 m with the current optics.

The uncertainty depends on several contributions. First of all the airglow layer height of 87 km is a statistical mean value, and may vary significantly. According to Baker and Stair (1988) a variation of ±4 km is adopted for its variability. Furthermore, the accuracy of the measurement setup is limited to ±0.5° concerning the zenith angle, and ±0.3 and ±0.2° concerning the aperture of the lens. This results in an overall uncertainty of ±400 km$^2$ or about 12 % for the covered area and an uncertainty of ±24 m per pixel, respectively. Since the uncertainty is dominated by the variability of the airglow layer height, these numbers are taken as a measure of precision, although the observational setup only limits the measurements' accuracy. If just considering the variability of the airglow layer the uncertainty for the visible area is 300 km$^2$ and consequently ±18 m per pixel.

However, for geographically mapping of the pixels, each of them is assigned a preliminary coordinate based on Eq. (1). A new equidistant grid is then constructed with a

**AMTD**

doi:10.5194/amt-2015-382

**A fast SWIR imager for observations of transient features in OH airglow**

P. Hannawald et al.

scale equal to the mean spatial resolution of 200 m and the preliminary coordinates are transformed to the new grid. This new grid is of the size of 320 to 306 pixel in the described setup. This results in some empty rows in the corrected image near the horizon where the available values are further apart, and more values available for one new grid point near the zenith where the original values are closer together. In the former case the missing values are interpolated by taking the mean value of up to eight non-empty nearest neighbours. In the latter case the mean of all available original values within the new grid point is calculated. Additionally to this mapping, the image is mirrored on the $y$ axis to change the view from a ground-based perspective to a satellite perspective. Figure 4a shows the raw (flat-field corrected) image. After the transformation and remapping we obtain the geographically corrected image (panel b). According to this, a wave field can easily be referred to a map (compare Fig. 4c).

In 2014 the instrument FAIM 1 was operated for 350 nights with the described setup. Measurements are performed for solar zenith angles larger than 96°. For each night, one row and one column of about 1000 images is taken and plotted versus time. These so-called keograms can easily be used to obtain information about cloudiness, incident moonlight or high atmospheric wave activity.

As a typical example, Fig. 5 shows keograms of the night from 3 to 4 October 2014. Between 17:30 and 00:40 UTC there are rather clear-sky conditions since stars are visible in the keograms, which is confirmed by the respective video sequences of this night. From 00:40 UTC onwards there are high-density clouds, which completely inhibit airglow observations.

Since FAIM 1 covers a rather broad spectral range from 950 to 1650 nm, several intercomparisons with co-located GRIPS-systems (Ground-Based Infrared P-Branch Spectrometer) have been performed. These instruments usually acquire airglow spectra between 1.5 and 1.6 µm, but they can be adjusted to record any other part of the airglow spectrum between approximately 0.9 and 1.65 µm. Usually, OH(3-1)-P-branch spectra acquired with these spectrometers are used to derive rotational temperatures with a temporal resolution of 5 s (GRIPS 13) or 15 s (GRIPS 16) (see Schmidt et al.,

Interactive Discussion

2013, for further details). For the investigation of the FAIM performance both spectrometers GRIPS 13 and GRIPS 16 were operated parallel to FAIM 1. The FOV of GRIPS 13 ($15° \times 15°$) is comparable in terms of size with the FOV of FAIM 1, whereas the FOV of GRIPS 16 is significantly smaller ($2° \times 2°$). In order to match the FOV of GRIPS 13,

5  the two-dimensional greyscale images of FAIM 1 are reduced to their mean value over the slightly smaller FOV of the spectrometer. On the other hand airglow spectra are integrated to yield one intensity value for each spectrum. Both time series are then averaged to get the same temporal resolution of 1 min. Since neither instrument was absolutely calibrated at this time, both time series have been normalized independently

10  to their individual maximum intensity.

Figure 6 shows the intensity time series, again for the night of 3 to 4 October 2014 until midnight, when clouds started to occur. The upper panel refers to the night from the beginning of the measurement, whereas the lower panel does not consider the first 25 min of twilight data. The time series appear to be anti-correlated during this time.

15  Therefore, the correlation coefficient increases from 0.15 to 0.87 when avoiding twilight. The discrepancy at dusk conditions is due to the emission of $O_2$(0-0) at 1.27 µm, which is decreasing exponentially after sunset (see e.g. Mulligan and Galligan, 1995).

Since the $O_2$(0-0) emission originates from different (variable) heights compared to OH and exhibits a rather long half-value time of approximately 1 h, the behaviour after

20  sunset is further investigated in order to estimate its impact on the observations. Hence, parallel measurements with FAIM 1, GRIPS 13 and GRIPS 16 have been performed, with GRIPS 13 adjusted to observe the 1.27 µm emission, and GRIPS 16 limited to the integrated OH(3-1)-Q-branch intensities (around 1.51 µm) to also avoid the weaker $O_2$(0-1) emission at 1.58 µm.

25  Figure 7 shows the different evolution of these three intensities normalised to their individual maximum. The start of each time series is marked with dashed lines. The $O_2$(0-0) intensity at 1.27 µm (black) shows the expected exponential decay after sunset, which is investigated in substantial detail by Mulligan and Galligan (1995), and almost no other small scale variation. The OH(3-1) intensity (blue) also shows the expected be-

**AMTD**

doi:10.5194/amt-2015-382

**A fast SWIR imager for observations of transient features in OH airglow**

P. Hannawald et al.

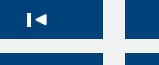
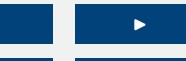
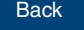
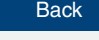

**AMTD**

doi:10.5194/amt-2015-382

**A fast SWIR imager for observations of transient features in OH airglow**

P. Hannawald et al.

haviour with rising intensities from 18:00 until 19:00 UTC, caused by increasing ozone concentrations, involved in the formation of excited OH. However, short period variations clearly dominate after 18:45 UTC. The evolution of the intensities recorded by FAIM 1 includes both oxygen and hydroxyl emissions. But the influence of OH – including a wide range of emissions between 0.9 and 1.65 μm – appears to be rather strong. After a sharp and short decrease in intensity directly after sunset, short periodic variations similar to the pure OH-emissions recorded by GRIPS 16 dominate the temporal evolution after 18:20 UTC. Obviously, the influence of the $O_2$-emission on the observation is of shorter duration than its half-value time of about 1 h. This result has been validated by using a longpass filter with 1260 nm cut-on wavelength in the FAIM setup (not shown). However, that does not improve the observations significantly, on the contrary it also excludes large portions of the OH-emissions. Therefore, it is not used in the regular observational setup anymore.

## 3  Data analysis

Data from the keograms of the night from 3 to 4 October 2014 shown in Fig. 5 are chosen for a case study. The time period of roughly one and a half hours between 21:05 and 22:30 UTC is investigated. The entire data set used for the analysis consists of 10 000 images in this time interval. To illustrate the wave structures, a series of images with a time difference of about four minutes between each of them is shown in Fig. 8. The series reveals an emerging wave structure (wave I) appearing and disappearing within about 20 min (21:40–22:00 UTC), clearly recognizable in the images (7)–(11). The black dashed line marks the approximate direction of the wave vector. This line is used as a transverse section through the images for further analysis.

A second wave (wave II) superposes wave (I) with a wave vector almost exactly perpendicular to wave (I). It is not as easily recognizeable in the images as the first wave, but Fig. 9 shows it without a doubt. The grey dotted line marks the direction of the wave vector of this wave (II). The superposition can best be seen in the images (7)–

(11). Images (9)–(11) show the presence of even smaller scale structures of less than 2 km. Similarly images (1)–(6) show a variety of different waves not further analysed here.

An intensity gradient is superimposed on each image, causing the upper part (low zenith angle) of the images to be darker than the lower (high zenith angle). This is due to the van-Rhijn-effect: the larger the zenith angle of the setup, the longer the line of sight through the airglow layer. This results in systematically higher recorded intensity. It is corrected with the formula derived by van Rhijn (1921) based on geometrical considerations, here used in the representation given by Roach and Gordon (1973):

$$I(\theta) = \frac{1}{\sqrt{1 - \left(\frac{R}{R+h}\right)^2 \cdot \sin^2(\theta)}},$$ (2)

with zenith angle $\theta \in (45° \pm 10°)$, earth radius $R$ (6371 km) and an airglow layer height $h$ of 87 km. The correction factor for each row of the original rectangular image is:

$$I_{\text{row,new}}(\theta) = I_{\text{row,old}} \cdot \frac{I(\theta)}{I(\gamma)},$$ (3)

where $I_{\text{row,new}}$ is the corrected (new) intensity of the original signal ($I_{\text{row,old}}$) recorded by one row of pixels and $\gamma$ is the zenith angle of the instrument's optical axis of 45° (see Fig. 2).

Further preprocessing includes removing the stars in the images to avoid a potential influence on later spectral analyses. They are identified by applying a gaussian blur on the image and subtracting this modified image from the original one. The gaussian blur is very sensitive for small structures with a strong intensity gradient – as is the case with stars. All pixels above a predefinded threshold will be treated as star pixels and further investigated to identify the radius of each star. For each star all pixels lying within this radius around the pixel with the highest intensity are identified as star pixels and removed from the original image.

# AMTD

doi:10.5194/amt-2015-382

**A fast SWIR imager for observations of transient features in OH airglow**

P. Hannawald et al.



Discussion Paper | Discussion Paper | Discussion Paper | Discussion Paper | Discussion Paper

Finally, the missing values for these pixels are interpolated over nearest neighbours of non-star pixels. The method works very well for bright stars. Some stars with an intensity only slightly above the background (airglow) intensity, resulting in an intensity below the threshold level, remain in the images, but do not have a large impact on the result of subsequent spectral analyses.

After these corrections the above-mentioned wave events are analysed by extracting data along the transverse sections shown in Fig. 8, obtaining 10 000 such sections for each wave. This is actually similar to taking a keogram for each propagation direction with a high temporal resolution.

Figure 9a and b depict the respective contour plots, with ordinates indicating the time in UTC and abscissas indicating the position along the transverse sections in kilometres (starting at the top part of the sections, compare Fig. 8). In order to determine the wavelengths from Fig. 9a and b some guiding lines are drawn along the apparent wavefronts. The distance between two of these lines in direction of the abscissa yields the wavelength and the ordinate distance yields the period of the wave structures. An uncertainty of ±2 km for the wavelengths and of ±100 s for the period is derived due to ambiguities of the guiding lines' positions.

Additionally, the wave parameters are also derived in Fourier space. Therefore, after removing linear trends and multiplication with a Hann window, the Fast Fourier Transformation (FFT) is applied for each time step shown in Fig. 9a and b.

The significance is derived by comparing 100 data sets with random numbers with the same mean and standard deviation as the original one, for each line. The insignificant values are indicated by white areas in the spectrograms Fig. 9c and d. It should be mentioned, that especially for larger wavelengths the uncertainty retrieved from the FFT will be higher, because of the discrete sampling bins and the calculation of the wavelength from the wavenumbers.

# AMTD

doi:10.5194/amt-2015-382

**A fast SWIR imager for observations of transient features in OH airglow**

P. Hannawald et al.

Discussion Paper | Discussion Paper | Discussion Paper | Discussion Paper |

Discussion Paper | Discussion Paper | Discussion Paper | Discussion Paper

**AMTD**

doi:10.5194/amt-2015-382

**A fast SWIR imager for observations of transient features in OH airglow**

P. Hannawald et al.

## 4   Results and discussion

In the night from 3 to 4 October 2014, two prominent wave events could be identified (see Fig. 8). For the smaller wave (I) a horizontal wavelength of about 6.8 km is determined on the basis of the wavenumber in the Fourier domain during the time interval from 21:36 to 21:54 UTC (see Fig. 9c). Its respective uncertainty amounts to ±0.4 km. Afterwards from 21:54 to 21:56 UTC the wavelength appears to shift to about 9.0 km (uncertainty range: 8.4–9.9 km) before it fades. If looking at the wave in position space instead (Fig. 9a), the wavelength can be determined to 7.0 km ± 2.0 km (with support of the guiding lines), the change in wavelengths however appears to be insignificant.

The main horizontal wavelength of wave (II), determined in the frequency domain, is 24.0 km (uncertainty range: 20–36 km, see Fig. 9d). Since the resolution for small wavenumbers is rather imprecise in Fourier space, its wavelength can be determined more precisely in position space (Fig. 9b), yielding in 20.2 km ± 2.0 km. Apparently, the uncertainty of the wavelength is smaller for wave (I) when analysed in the frequency domain and vice versa for wave (II). The values are summarised in Table 1.

While wave (I) can be observed for about 20 min, wave (II) is apparent in the Fourier spectrum (Fig. 9d) for about 35 min from 21:40 until 22:15 UTC. The Fourier amplitude maximises at 21:51 UTC, just when wave (I) is starting to diminish.

The period of wave (I) amounts to 1400 s ± 100 s determined by the ordinate distance the guiding lines in Fig. 9a. Wave (II) has a period of 585 s ± 100 s (see Fig. 9b).

Thus, the phase velocity can be calculated according to the following equation:

$$v_{\text{phase,horizontal}} = \frac{\lambda}{T} \pm \left( \frac{\Delta\lambda}{T} + \frac{\lambda}{T^2} \cdot \Delta T \right), \tag{4}$$

with the period $T \pm \Delta T$ and the horizontal wavelength $\lambda \pm \Delta\lambda$. $\Delta\lambda$ is either the uncertainty of 2.0 km in position space (referring to $\lambda_1$) or half the size of the specific uncertainty range (referring to $\lambda_2$). As discussed before, the analysis in frequency domain is more suitable for wave (I), whereas for wave (II) the analysis in position space exhibits

lower uncertainty. Considering this, wave (I) is propagating with a phase velocity of $4.9\,\mathrm{m\,s^{-1}} \pm 0.6\,\mathrm{m\,s^{-1}}$ and wave (II) with $34.5\,\mathrm{m\,s^{-1}} \pm 9.3\,\mathrm{m\,s^{-1}}$.

We speculate that the small wavelength, low lifetime and the perpendicular direction of propagation may indicate a ripple structure as defined by Adams et al. (1988) and Taylor and Hapgood (1990). They are distinguished from larger scale structures termed as bands. Ripples are strongly related to Kelvin–Helmholtz-instabilities (KHI) and convective instabilities (Hecht, 2004). Images (9)–(11) of Fig. 8 show further small scale features, which we assume to be KHI-billows. Fritts et al. (2014) show structures based on model calculations of the OH airglow response to KHIs looking very similar to these. Yamada et al. (2001) and Hecht et al. (2014) present similar phenomena in their measurements. The latter ones provide a detailed analysis combining the measurements of different instruments.

In order to investigate potential influences of these small scale waves on mesopause temperatures, parallel measurements of rotational temperatures taken with GRIPS 13 are shown in Fig. 10. The top panel shows the 1 min moving average of the OH rotational temperature time series (resolution: 5 s). The mean temperature of 205.7 K is indicated by the dashed horizontal line. The thick smooth line represents a spline of the time series to reveal the underlying long-period structure (filtering periods lower than about 200 s). The Brunt–Väisälä-period is calculated with the given temperature for vertical temperature gradients between zero and two kelvins per kilometre being between 272 and 299 s (gravity acceleration is $9.54\,\mathrm{m\,s^{-1}}$ in 87 km).

The bottom panel in Fig. 10 shows the Fourier transform of the time series, the dashed line gives the 0.95-confidence level calculated for each frequency based on 10 000 time series of random data. Significant periods beyond the Brunt–Väisälä-period are around 2500, 1000, 510 and 390 s. The period of 510 s lies within the uncertainty range of wave (II) observed in the FAIM images (585 s ± 100 s). Furthermore, it also maximises at 21:55 UTC (compare thick line in Fig. 10). The period of wave (I) is not clearly visible in the GRIPS time series due to the FOV (around 47 km × 35 km), which cannot resolve the small wavelength of wave (I). However, the 1000 s period may be

**AMTD**

doi:10.5194/amt-2015-382

**A fast SWIR imager for observations of transient features in OH airglow**

P. Hannawald et al.

Interactive Discussion

tentatively interpreted as a residual signal caused by this wave. The 40 min period can actually be identified in the FAIM images in Fig. 9a, with wavefronts in horizontal direction. Overall, the temperature appears to decrease by a few Kelvin after 22:00 UTC, which is after the maximum recorded amplitudes of the wave structures and their following disappearance.

## 5 Summary and conclusions

We developed the new airglow imager FAIM 1 based on an InGaAs detector, sensitive to the bright OH-emissions between 900 and 1650 nm. Thus, two frames per second can be acquired at a spatial resolution of 200 m with current optics. Important features of the instrument, especially a noise level of only 25 counts for by a sensor temperature of 235 K, are determined, while signal levels are typically around 850.

The processing chain, e.g., geographical correction of the images for the standard setup with 45° zenith angle are presented. The data of FAIM 1 is compared to GRIPS airglow spectrometric observations (Schmidt et al., 2013). In comparison with two such GRIPS instruments, one recording the 1.27 µm $O_2$(0-0)-emission and one recording the bright OH(3-1)-and OH(4-2)-emissions, it was shown, that the $O_2$ is dominating the FAIM 1 data only during a rather short period of time after sunset. It is worth noting, that this time period is shorter than the chemical lifetime of 1 h of the excited $O_2$ state. During clear sky conditions the broadband FAIM 1 data show a high correlation of up to 0.99 with the spectrally resolved GRIPS data.

A case study was performed in order to demonstrate the capability of the instrument to observe smaller scale gravity wave structures in the OH airglow layer in about 87 km altitude. During the night from 3 to 4 October 2014 two prominent wave structures were identified and analysed. A smaller structure with about 7 km horizontal wavelength is probably part of a dissipation process of a larger one with about 20 km horizontal wavelength. The small wave has a nearly perpendicular direction of propagation to

**AMTD**

doi:10.5194/amt-2015-382

**A fast SWIR imager for observations of transient features in OH airglow**

P. Hannawald et al.

Discussion Paper | Discussion Paper | Discussion Paper | Discussion Paper

the larger one and a short lifetime of 20 min. It is therefore tentatively interpreted as a so-called ripple structure.

Where the superposition of both waves takes place, one can see even smaller structures of the order of about 2 km, which we assume to be Kelvin–Helmholtz-Instability billows (compare Fritts et al., 2014; Hecht et al., 2014).

In the FAIM 1 data of 2014 there are more examples for billow and ripple phenomena which are not yet analysed in such great detail. It is an open question whether these phenomena are actually common or the instrument's setup and site offer a unique possibility to study them. Further investigations and statistics of more nights and from other sites may help answering this question.

*Acknowledgements.* This work is funded by the Bavarian State Ministry of the Environment and Consumer Protection by grant no. TUS01UFS-67093. The investigated data is archived at WDC-RSAT (World Data Center for Remote Sensing of the Atmosphere). The observations are part of NDMC (http://wdc.dlr.de/ndmc).

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

Title Page

Abstract | Introduction

Conclusions | References

Tables | Figures

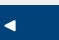 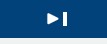

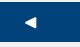 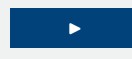



**Table 1.** Summary of the determined wave parameters identified in Fig. 9. Wave structure (I) is separated into (I.1), lasting until 21:54 UTC and (I.2), emerging at 21:54 UTC. $\lambda$ is the horizontal wavelength once determined in position space $\lambda_1$ (see Fig. 9a and b, respectively), once in frequency domain $\lambda_2$ (panel c and d). $\Delta t$ is the lifetime determined in the frequency domain (panel c and d), $T$ is the wave period (determined from panel a and b) and $v$ the horizontal phase velocity referring to $\lambda_1$ and $\lambda_2$.

| | $\lambda_1$ (km) | $\lambda_2$ (km) | $\Delta t$ (min) | $T$ (s) | $v_1$ (m s$^{-1}$) | $v_2$ (m s$^{-1}$) |
|---|---|---|---|---|---|---|
| wave (I.1) | $7.0 \pm 2$ | 6.8 (6.4–7.2) | 18 | $1400 \pm 100$ | $5.0 \pm 1.8$ | $4.9 \pm 0.6$ |
| wave (I.2) | – | 9.0(8.4–9.9) | 2 | – | – | – |
| wave (II) | $20.2 \pm 2$ | 24 (20–36) | 35 | $585 \pm 100$ | $34.5 \pm 9.3$ | $39.3 \pm 21$ |

Discussion Paper | Discussion Paper | Discussion Paper | Discussion Paper

Discussion Paper | Discussion Paper | Discussion Paper | Discussion Paper | Discussion Paper |

**AMTD**

doi:10.5194/amt-2015-382

**A fast SWIR imager for observations of transient features in OH airglow**

P. Hannawald et al.



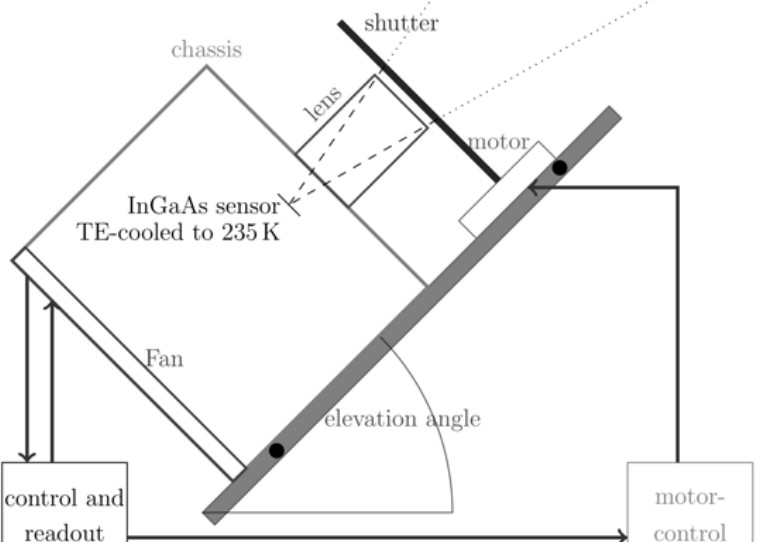

**Figure 1.** Sketch of the instrument and measurement setup. See text for further details.

Discussion Paper | Discussion Paper | Discussion Paper | Discussion Paper |

**AMTD**

doi:10.5194/amt-2015-382

**A fast SWIR imager for observations of transient features in OH airglow**

P. Hannawald et al.

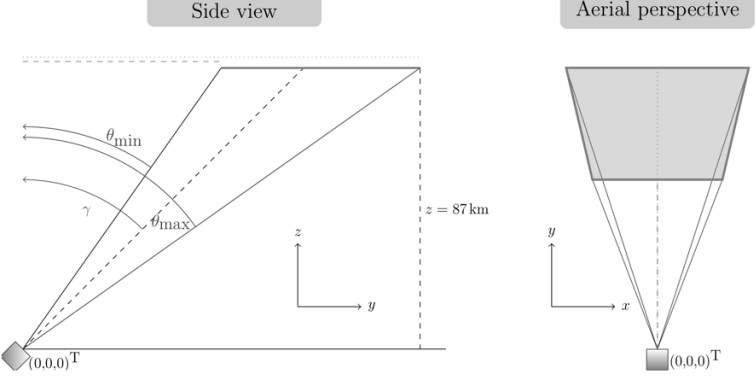

**Figure 2.** Measurement setup with a zenith angle of 45° viewed from the side and from aerial perspective. The axis show the definition of the coordinate system used to reference the pixels to geographical coordinates. The instrument is located at the center of the coordinate system, $z = 87$ km is the altitude of the OH airglow layer, $\gamma$ is the zenith angle of the instrument and $\theta$ is the variable in the range of $\gamma \pm \mathrm{FOV}_{\mathrm{vertical}}$, FOV being the aperature angle of the instrument of 10° (vertical) and 12° (horizontal).

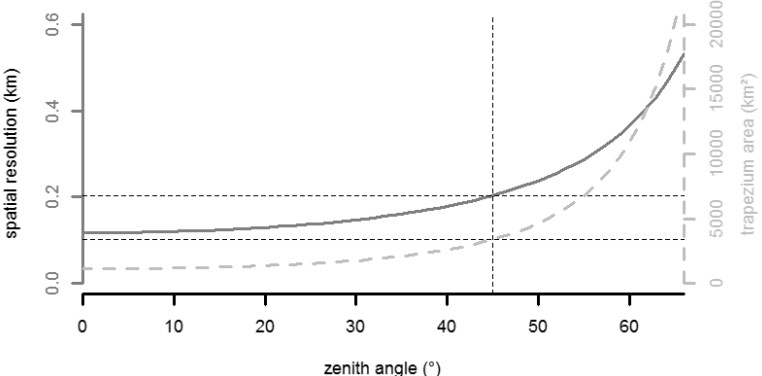

**Figure 3.** Mean spatial resolution of the acquired images as a function of zenith angle (black). The grey dashed line shows the respective area of the trapezium shaped FOV with the axis on the right side of the graph. The standard angle of 45° is marked by thin dashed lines for both curves showing a mean spatial resolution of 200 m and a corresponding area of 3400 km².

Discussion Paper | Discussion Paper | Discussion Paper | Discussion Paper |

**AMTD**

doi:10.5194/amt-2015-382

**A fast SWIR imager for observations of transient features in OH airglow**

P. Hannawald et al.



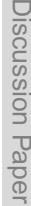

**AMTD**

doi:10.5194/amt-2015-382

A fast SWIR imager
for observations of
transient features in
OH airglow

P. Hannawald et al.

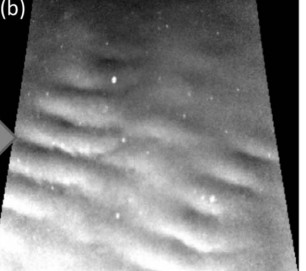

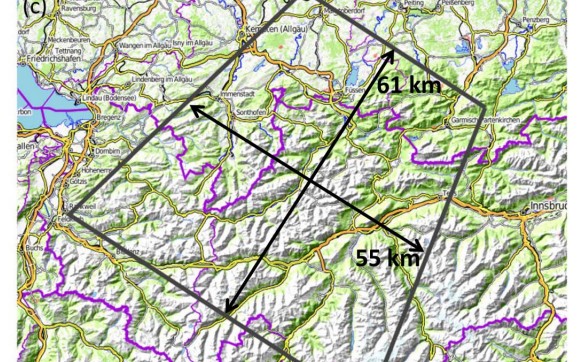

**Figure 4. (a)** shows the flat-field corrected image which is distorted due to the geometry of the measurement setup with a zenith angle of 45°. In **(b)** this distortion is corrected by Eq. (1). Additionally, the image is mirrored on the middle axis to change the ground-based observer's view to a satellite perspective. **(c)** shows the position of the FOV within the Alpine region (source: www.opentopomap.org, Oktober 2015).

**AMTD**

doi:10.5194/amt-2015-382

A fast SWIR imager
for observations of
transient features in
OH airglow

P. Hannawald et al.

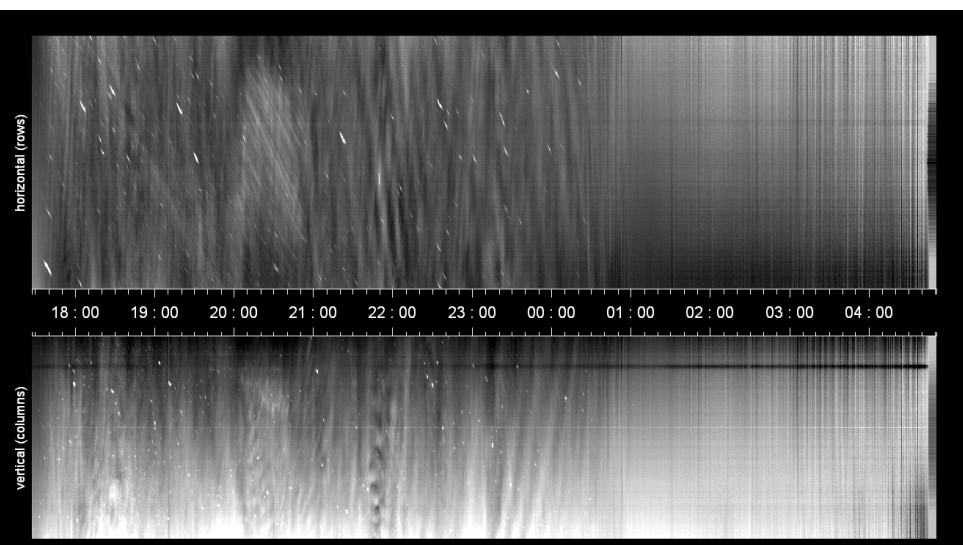

**Figure 5.** Keogram for one row and one column of images for the night of 3 to 4 October 2014. Between 17:30 and 00:40 UTC stars can be seen clearly, but after 00:40 UTC dense cloud cover appears. Several wave events can easily be identified at 18:30, 20:00–21:00, 22:00 and 23:30 until 00:30 UTC. The time interval between 21:05 and 22:30 UTC is further investigated in Sects. 3 and 4.

Discussion Paper | Discussion Paper | Discussion Paper | Discussion Paper

**AMTD**

doi:10.5194/amt-2015-382

**A fast SWIR imager for observations of transient features in OH airglow**

P. Hannawald et al.



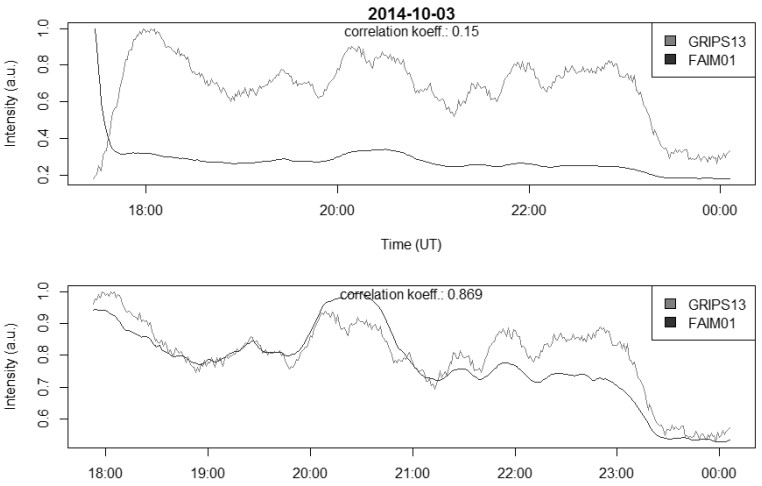

**Figure 6.** Intensity time series for the night of 3 to 4 October 2014. The black line shows the mean value over all pixels of FAIM 1 within the FOV of GRIPS 13 and the grey line shows the intensity measured with GRIPS 13 integrated between 1500 and 1600 nm. Both data sets are averaged to 1 min mean values to avoid effects of different sampling rates and normalised to their individual maximum. Top: the entire time interval from the start of the measurement until 00:00 UTC, when clouds emerged. A correlation of only 0.15 is determined. Bottom: same as before, but the first 25 min of twilight are avoided. The correlation now increases to 0.87.

**AMTD**

doi:10.5194/amt-2015-382

A fast SWIR imager
for observations of
transient features in
OH airglow

P. Hannawald et al.

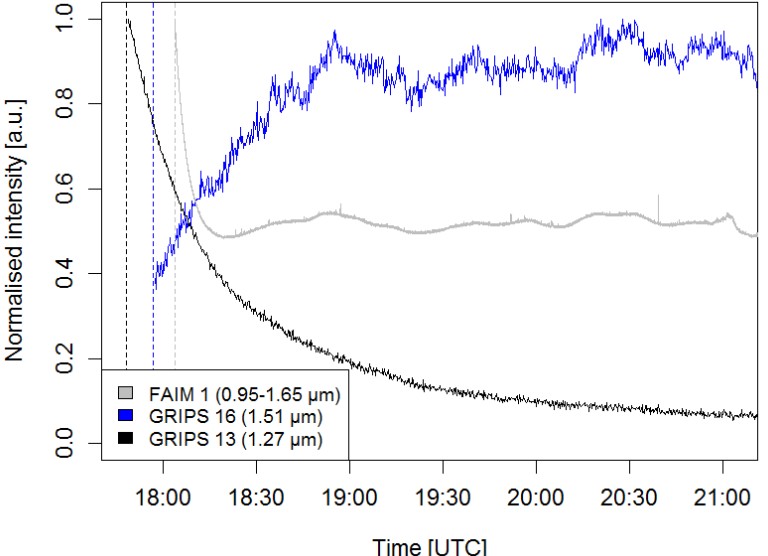

**Figure 7.** Parallel measurements of FAIM 1 with spectral range from 950 to 1650 nm, GRIPS 13 adjusted for measuring 1.27 µm $O_2$(0-0)-transition and GRIPS 16 for the OH(3-1)-Q-branch-transition at 1.51 µm for the first 3 h of the night. The case study shows an exponential decay of $O_2$ and an increase of OH intensity. The FAIM time series (averaged over the FOV of GRIPS 13) shows a mixture of both behaviours.

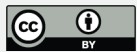

(1) 2014-10-03 21:05:00
(2) 2014-10-03 21:10:39
(3) 2014-10-03 21:16:19
(4) 2014-10-03 21:22:00
(5) 2014-10-03 21:27:39
(6) 2014-10-03 21:33:19
(7) 2014-10-03 21:39:00
(8) 2014-10-03 21:44:39
(9) 2014-10-03 21:50:19
(10) 2014-10-03 21:56:00
(11) 2014-10-03 22:01:39
(12) 2014-10-03 22:07:20
(13) 2014-10-03 22:13:00
(14) 2014-10-03 22:18:39
(15) 2014-10-03 22:24:20
(16) 2014-10-03 22:30:00

N

(km)
0  5  10 15

**AMTD**

doi:10.5194/amt-2015-382

**A fast SWIR imager for observations of transient features in OH airglow**

P. Hannawald et al.

**Figure 8.** Series of consecutive images within the chosen time interval from 21:05 to 22:30 UTC. The black dashed line indicates the transverse section through the images used to analyse the smaller scale wave (I). The grey dotted line shows a transverse section approximately in direction of the wave vector for the investigation of a faint larger wave structure (II) almost perpendicular to the first one. The whole observed area is about 55 km (central width of the FOV) times 60 km (height of the FOV).

**AMTD**

doi:10.5194/amt-2015-382

**A fast SWIR imager for observations of transient features in OH airglow**

P. Hannawald et al.

# AMTD

doi:10.5194/amt-2015-382

**A fast SWIR imager for observations of transient features in OH airglow**

P. Hannawald et al.

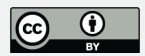

**Figure 9. (a)** and **(b)** show the temporal evolution of the transverse sections of waves (I) and (II). The lines indicate the position of the wavefronts. The ascissa corresponds to the position within the transverse sections shown in Fig. 8 (axis origin corresponds to the upper part of the transverse sections). Panels **(c)** and **(d)** show the Fourier transforms of each line of **(a)** and **(b)**, respectively. The white areas are not significant on a 95 %-level.

**AMTD**

doi:10.5194/amt-2015-382

**A fast SWIR imager for observations of transient features in OH airglow**

P. Hannawald et al.

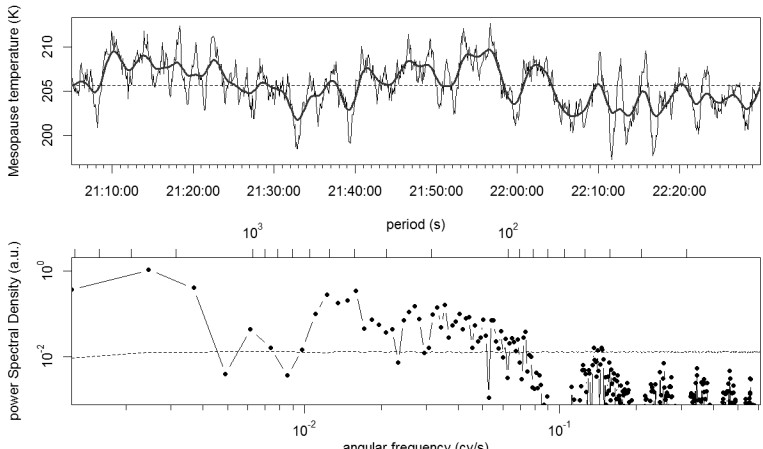

**Figure 10.** 1 min moving averages of the time series of GRIPS 13 looking into the same direction as FAIM 1. The dashed line shows the mean value for the chosen time interval. A spline is drawn as thick line to guide the eye. The bottom panel shows the Fourier transform of the time series. The dashed line gives the 95 %-confidence level per frequency based on 10 000 time series of random data.

**AMTD**

doi:10.5194/amt-2015-382

**A fast SWIR imager for observations of transient features in OH airglow**

P. Hannawald et al.