# Peer review of "Atmos. Meas. Tech. Discuss., doi:10.5194/amt-2015-382, 2016 Manuscript under review for journal Atmos. Meas. Tech. Published: 14 January 2016 © Author(s) 2016. CC-BY 3.0 License."

_Atmospheric Measurement Techniques, 2015_

## Referee Comment (RC1) · Anonymous Referee #1 · 29 Jan 2016

Journal: AMT Title: A fast SWIR imager for observations of transient features in OH airglow Author(s): P. Hannawald et al. MS No.: amt-2015-382 MS Type: Research article

This paper describes a new type of narrow-field IR imaging system called the Fast Airglow IMager 1 (FAIM 1). The imager has been designed to record atmospheric gravity waves in the mesospheric hydroxyl nightglow from the 85 km altitude region. In particular, the system yields high signal-to-noise images of OH gravity waves at ∼0.5s cadence and so can provide high time-resolution of local wave dynamics.

The paper begins with a discussion of the FAIM design and its field operation. A description is then made of the preliminary reduction and analysis of the data acquired by the by FAIM instrument using a pilot set of data. The results are shown and compared

to temperature measurements obtained by a co-located spectrograph (GRIPS).

A minor comment would be that the comparison of the FAIM images and the GRIPS 13 temperatures on Pages 13-14 and in Figure 10 is poorly described and vague. The description needs to be clarified. Otherwise, I recommend that the paper be published.

---

## Referee Comment (RC2) · Anonymous Referee #2 · 18 Feb 2016

This manuscript describes a new instrument setup to observe at a fast rate (images every ~0.5s) small scale structures visible in the OH airglow emission, at ~87km. Even if airglow imagers are not new, this setup might be interesting to investigate ripple-like features. The authors carefully looked at the impact of the O2(0,0) emission on their measurements and also compared their results with a separate instrument (GRIPS). The wave analysis technique is explained with a couple of examples. A complete analysis of their dataset should provide interesting results on the occurrence and characteristics of instability features in the MLT. Nevertheless, for a complete setup, I would suggest they also use a large FOV imager (maybe all-sky with a similar detector) to assess the atmospheric context.

Minor points: p2 l.21: a unique l.23: Atmospheric gravity waves especially are... p3 l.8: most lidars have much better vertical resolution than radars! p4 l.1: using narrowband

[Figure]

filters... l.6: in a parallel direction l.15: The 320x256 pixel (or "by" instead of "x") l.16: based on InGaAs technology... l.23: are used. p5 l.3: The airglow signal l.5: images with the same exposure time... l.12: which is neglected. l.14: at the altitude of the OH emission peak, at 87km... l.19: The observed trapezium-shaped area of the airglow layer is the projection of the rectangular-shaped sensor due to the observation geometry. p6 l.4: (see Fig. 2)... at constant altitude... l.12: The standard... l.16: ...(1988) the variation is +/-4km. l.25: mapping the pixels,... p7 l.1-12: seems complicated, maybe calculating the positions in reverse would be easier l.18: keograms for the night... l.24: no need to write again what GRIPS means p8 l.12: started to appear. p9 l.24: superimposes on wave (I)... p10 l.19-20: jumped to the next line l.21: predefined p12 l.8: If measuring the wave parameters from the projected image (Fig. 9a),... l.15: but it is the opposite for wave (II). p13 l.3: short lifetime l.18: ...series revealing... l.19-21: The Brunt-Vaisala period calculated for the given temperature and for vertical gradients between 0 and 2K.km-1 varies between 272 and 299s... l.21: at 87km p14 l.10: for a sensor... l.17: remove comma after "noting" l.20: didn't say that in the text (0.99) l.22: at about 87km l.26: propagation compared to...

Figure 3: left side axis should be in (km/pixel) Figure 4c: could be a little be larger to include the complete FOV (bit missing at the bottom)

[Figure]

---

## Author Comment (AC1) · 4 Mar 2016

The authors would like to thank Anonymous Referee #1 for the comment (29 January 2016), especially for finding missing details within the manuscript regarding the analysis of the GRIPS 13 time series.

Anonymous Referee #1 commented "that the comparison of the FAIM images and the GRIPS 13 temperatures on Pages 13-14 and in Figure 10 is poorly described and vague. The description needs to be clarified."

In respond to the comment, we changed the paragraphs beginning at p13, I.13 and ending on p14, I.5. The changes belong to a clearification of the parallel direction of measurements of the instruments GRIPS and FAIM during this specific time span. We changed the degrees of freedom of the spline to better visualise a long-period Printer-friendly version

temperature oscillation and want to mention that the spline is just to guide the eye. We decided that the calculation of the Brunt-Väisälä-period is unnecessary here and omitted it accordingly.

We added some detail concerning the identification of significant periods within the series (by comparison with a set of random numbers) and point out the phenomenon of observational selection, meaning that waves with small wavelengths are more likely to be visible in the imager data (FAIM) and waves with large wavelengths are not visible in the images analysed with the methods presented. The description of figure 10 is rephrased and references to the text for further details.

The authors hope that the paragraphs and the figure description is now formulated in a clearer manner.

**1.1 New Version**

In order to investigate potential influences of these small scale waves on mesopause temperatures, the upper panel of Fig. 10 shows the variation of rotational temperatures observed with GRIPS 13 at the same time shown as one minute running mean values and a original resolution of 5 s. During this time span the instrument measured in parallel direction with FAIM 1. The mean temperature in the observed time range is 205.7 K. The thick smooth line represents a spline of the time series revealing the underlying long-period structure (filtering periods lower than about 1000 s) and is included to guide the eye. Obviously, two wave crests can be seen in the temperature, dissolving after 22:10 UTC.

The bottom panel of Fig. 10 shows the Fourier transform of the time series and the dashed line gives the 0.95-confidence level calculated for each frequency based on 10000 random number time series with same mean and standard deviation as the original time series. Significant oscillations are identified at periods of 2500 s, 1000 s,
510 s and 390 s. The periods found in the FAIM images and the periods of the GRIPS temperatures are difficult to match. A reason for that is observational selection, meaning that, on the one hand, the small wave structure (I) in the images has several wave crests and troughs within the FOV of GRIPS, which cannot resolve it properly because its spatial resolution is similar to the size of the whole FOV of FAIM. On the other hand, the period of 510 s found in the temperature data of GRIPS lies within the uncertainty range of wave (II) observed in the FAIM images ( $585 s \pm 100 s$ ). This wave structure has a wavelength of about 20 km and should be resolved by the GRIPS instrument. Other periods may correspond to larger scale waves which are not visible in the FAIM images itself. However, the disappearing temperature oscillation can tentatively be interpreted as a breakdown of a larger scale wave into smaller structures, which are clearly visible in the FAIM images and probably decay into KHI.

**1.1.1 Fig 10: new caption**

Upper panel: One minute running averages of the time series of GRIPS 13 measuring in parallel direction with FAIM 1. A smooth spline is drawn as thick line to guide the eye and show a long-period wave structure which dissolves at the end of the shown interval. Bottom panel: Fourier transform of the temperature time series. Periods of 2500, 1000, 510 and 390 s are visible in the range of higher periods. The dashed line gives the 0.95 significance. See text for further details.

**1.2 Old Version**

In order to investigate potential influences of these small scale waves on mesopause temperatures, parallel measurements of rotational temperatures taken with GRIPS 13 are shown in Fig. 10. The top panel shows the one minute moving average of the OH rotational temperature time series (resolution: 5 seconds). The mean temperature of
205.7 K is indicated by the dashed horizontal line. The thick smooth line represents a spline of the time series to reveal the underlying long-period structure (filtering periods lower than about 200 s). The Brunt-Väisälä-period is calculated with the given temperature for vertical temperature gradients between zero and two kelvins per kilometre being between 272 s to 299 s (gravity acceleration is  $9.54 \frac{m}{s^2}$  in 87 km).

The bottom panel in Fig. 10 shows the Fourier transform of the time series, the dashed line gives the 0.95-confidence level calculated for each frequency based on 10,000 time series of random data. Significant periods beyond the Brunt-Väisälä-period are around 2500 s, 1000 s, 510 s and 390 s. The period of 510 s lies within the uncertainty range of wave (II) observed in the FAIM images ( $585 s \pm 100 s$ ). Furthermore, it also maximises at 21:55 UTC (compare thick line in Fig. 10). The period of wave (I) is not clearly visible in the GRIPS time series due to the FOV (around  $47 \text{ km} \times 35 \text{ km}$ ), which cannot resolve the small wavelength of wave (I). However, the 1000 s period may be tentatively interpreted as a residual signal caused by this wave. The 40 minutes period can actually be identified in the FAIM images in Fig. 10 (a), with wavefronts in horizontal direction. Overall, the temperature appears to decrease by a few Kelvin after 22:00 UTC, which is after the maximum recorded amplitudes of the wave structures and their following disappearance.
Fig. 1. new caption: see text

---

## Author Comment (AC2) · 4 Mar 2016

The authors would like to thank anonymous referee #2 for the comments (18 February 2016). We also thank the referee for the note concerning the transformation algorithm. In respond to the comment, we changed most of the minor points exactly as suggested by the referee. A few points were formulated in a slightly different way. A detailed list containing all comments can be found below and the exact changes within the manuscript can be found in the attached marked-up manuscript version.

- Minor changes on p2 l.21 and l.23 were applied.

- p3 l.8: The vertical resolution of radars and lidars is clearified according to the referee's comment. Original: " Radars and lidars for example measure changes

in temperature and wind with a comparatively high vertical resolution...", Changed to: "Radars and especially lidars for example measure changes in temperature and wind with a comparatively high vertical resolution...")

- p4 l.1, l.6 (also changed in description of Fig. 7), l.15, l.16 and l.23 are corrected. The wording of p5 l.3 is changed, the comment of l.5 is added, and l.12 "furthermore" removed. The prepositions at l.14 are corrected and l.19 is now formulated clearer according to the comment. p6 l.4 both corrected. l.12 and l.16 corrected.

- p7 l.1-12 Thank you for your comment. When we started to develop the transformation algorithm it seemed more reasonable to start at the sensor instead of the airglow layer, but we will think about that point for future developments.

- p7 l.18, l.24, p8 l.12 and p9 l.24 are corrected.

- p10 l.19-20 we could not find a line jump in the source, but it obviously appeared in the manuscript. We think that this has to be checked in the final step of the review process.

- p12 l.8: changed the mentioned line in a different way then suggested( Original: "If looking at the wave in position space instead (Fig. 9 (a)), the wavelength can be determined to 7.0 km $\pm$ 2.0 km (with support of the guiding lines), ..." Changed to: "If considering the transverse sections of wave (I) instead (Fig. 9 (a)), the wavelength can be determined to 7.0 km $\pm$ 2.0 km (with support of the guiding lines), ..."). Measuring the wave from the projected images is possible, but should then better refer to Fig. 8.

- p12 l.15, p13 l.3 and l.18 are changed according to the comments.

- l.19-21 are obsolete with the rephrasing in respond to anonymous referee #1 who wished a clearification of this paragraph. The calculation of the Brunt-Väisälä-period was decided not to be important in this context any more.

- p14 l.10, l.17 corrected.

- p14 l.20 We forgot to add the information about investigating other nights with even higher correlation coefficients up to 0.99 in the text. Thank you very much for mentioning that.

- l.22 and l.26 corrected. Fig. 3 axis now has the correct unit (km/pixel) and Fig 4c. now includes the complete FOV.

Please also note the supplement to this comment:
http://www.atmos-meas-tech-discuss.net/amt-2015-382/amt-2015-382-AC2-supplement.pdf

[Figure]

**Fig. 1.**

[Figure]

**Fig. 2.**

**Supplement:**

Manuscript prepared for Atmos. Meas. Tech.
with version 2015/04/24 7.83 Copernicus papers of the LaTeX class copernicus.cls.
Date: 4 March 2016

**A fast SWIR imager for observations of transient features in OH airglow**

Patrick Hannawald[1], Carsten Schmidt[2], Sabine Wüst[2], and Michael Bittner[1,2]

[1]University of Augsburg, Germany - Institute of Physics
[2]German Aerospace Center, Germany - German Remote Sensing Data Center

*Correspondence to:* Patrick Hannawald (patrick.hannawald@physik.uni-augsburg.de)

[revised manuscript text omitted]
 to 9.9 km) before it fades. If  considering the transverse sections of wave (I) instead (Fig. 9 (a)), the wavelength can be determined to 7.0 km ± 2.0 km (with support of the guiding lines), the change in wavelengths however appears to be insignificant.

The main horizontal wavelength of wave (II), determined in the frequency domain, is 24.0 km (uncertainty range: 20 km to 36 km, see Fig. 9 (d)). Since the resolution for small wavenumbers is rather imprecise in Fourier space, its wavelength can be determined more precisely in position space (Fig. 9 (b)), yielding in 20.2 km ± 2.0 km. Apparently, the uncertainty of the wavelength is smaller for wave (I) when analysed in the frequency domain, but it is the opposite for wave (II). The values are summarised in table 1.

While wave (I) can be observed for about 20 min, wave (II) is apparent in the Fourier spectrum (Fig. 9 (d)) for about 35 min from 21:40 UTC until 22:15 UTC. The Fourier amplitude maximises at 21:51 UTC, just when wave (I) is starting to diminish.

The period of wave (I) amounts to 1400 s ± 100 s determined by the ordinate distance the guiding lines in 9 (a). Wave (II) has a period of 585 s ± 100 s (see Fig. 9 (b)).

Thus, the phase velocity can be calculated according to the following equation:

$$v_{\text{phase,horizontal}} = \frac{\lambda}{T} \pm \left( \frac{\Delta\lambda}{T} + \frac{\lambda}{T^2} \cdot \Delta T \right), \tag{4}$$

with the period $T \pm \Delta T$ and the horizontal wavelength $\lambda \pm \Delta\lambda$. $\Delta\lambda$ is either the uncertainty of 2.0 km in position space (referring to $\lambda_1$) or half the size of the specific uncertainty range (referring to $\lambda_2$). As discussed before, the analysis in frequency domain is more suitable for wave (I), whereas for wave (II) the analysis in position space exhibits lower uncertainty. Considering this, wave (I) is propagating with a phase velocity of 4.9 $\frac{\text{m}}{\text{s}}$ ± 0.6 $\frac{\text{m}}{\text{s}}$ and wave (II) with 34.5 $\frac{\text{m}}{\text{s}}$ ± 9.3 $\frac{\text{m}}{\text{s}}$.

We speculate that the small wavelength,  short lifetime and the perpendicular direction of propagation may indicate a ripple structure as defined by Adams et al. (1988) and Taylor and Hapgood (1990). They are distinguished from larger scale structures termed as bands. Ripples are strongly related to Kelvin-Helmholtz-instabilities (KHI) and convective instabilities (Hecht (2004)). Images (9) to (11) of Fig. 8 show further small scale features, which we assume to be KHI-billows. Fritts et al. (2014) show structures based on model calculations of the OH airglow response to KHIs looking very similar to these. Yamada et al. (2001) and Hecht et al. (2014) present similar phenomena in their measurements. The latter ones provide a detailed analysis combining the measurements of different instruments.

In order to investigate potential influences of these small scale waves on mesopause temperatures, the upper panel of Fig. 10 shows the variation of rotational temperatures observed with GRIPS 13 at the same time shown as one minute running mean values and a original resolution of 5 s. During this time span the instrument measured in parallel direction with FAIM 1. The mean temperature in the observed time range is 205.7 K. The thick smooth line represents a spline of the time series revealing the underlying long-period structure (filtering periods lower than about 1000 s) and is included to guide the eye. Obviously, two wave crests can be seen in the temperature, dissolving after 22:10 UTC.

The bottom panel of Fig. 10 shows the Fourier transform of the time series and the dashed line gives the 0.95-confidence level calculated for each frequency based on 10000 random number time series with same mean and standard deviation as the original time series. Significant oscillations are identified at periods of 2500 s, 1000 s, 510 s and 390 s. The periods found in the FAIM images and the periods of the GRIPS temperatures are difficult to match. A reason for that is observational selection, meaning that, on the one hand, the small wave structure (I) in the images has several wave crests and troughs within the FOV of GRIPS, which cannot resolve it properly because its spatial resolution is similar to the size of the whole FOV of FAIM. On the other hand, the period of 510 s found in the temperature data of GRIPS lies within the uncertainty range of wave (II) observed in the FAIM images (585 s$\pm$100 s). This wave structure has a wavelength of about 20 km and should be resolved by the GRIPS instrument. Other periods may correspond to larger scale waves which are not visible in the FAIM images itself. However, the disappearing temperature oscillation can tentatively be interpreted as a breakdown of a larger scale wave into smaller structures, which are clearly visible in the FAIM images and probably decay into KHI.

**5 Summary and conclusions**

We developed the new airglow imager FAIM 1 based on an InGaAs detector, sensitive to the bright OH-emissions between 900 nm and 1650 nm. Thus, two frames per second can be acquired at a spatial resolution of 200 m with current optics. Important features of the instrument, especially a noise level of only 25 Counts for  a sensor temperature of 235 K, are determined, while signal levels are typically around 850.

The processing chain, e.g., geographical correction of the images for the standard setup with 45° zenith angle are presented. The data of FAIM 1 is compared to GRIPS airglow spectrometric observations Schmidt et al. (2013)). In comparison with two such GRIPS instruments, one recording the $1.27\,\mu m$ $O_2$(0-0)-emission and one recording the bright OH(3-1)-and OH(4-2)-emissions, it was shown, that the $O_2$ is dominating the FAIM 1 data only during a rather short period of time after sunset. It's worth noting  that this time period is shorter than the chemical lifetime of one hour of the excited $O_2$ state. During clear sky conditions the broadband FAIM 1 data show a high correlation of up to 0.99 with the spectrally resolved GRIPS data.

A case study was performed in order to demonstrate the capability of the instrument to observe smaller scale gravity wave structures in the OH airglow layer  at about 87 km altitude. During the night from 3rd to 4th October 2014 two prominent wave structures were identified and analysed. A smaller structure with about 7 km horizontal wavelength is probably part of a dissipation process of a larger one with about 20 km horizontal wavelength. The small wave has a nearly perpendicular direction of propagation compared to the larger one and a short lifetime of 20 minutes. It is therefore tentatively interpreted as a so-called ripple structure.

Where the superposition of both waves takes place, one can see even smaller structures of the order of about 2 km, which we assume to be Kelvin-Helmholtz-Instability billows (compare Fritts et al. (2014), Hecht et al. (2014)).

In the FAIM 1 data of 2014 there are more examples for billow and ripple phenomena which are not yet analysed in such great detail. It is an open question whether these phenomena are actually common or the instrument's setup and site offer a unique possibility to study them. Further investigations and statistics of more nights and from other sites may help answering this question.

*Acknowledgements.* This work is funded by the Bavarian State Ministry of the Environment and Consumer Protection by grant number TUS01UFS-67093. The investigated data is archived at WDC-RSAT (World Data Center for Remote Sensing of the Atmosphere). The observations are part of NDMC (http://wdc.dlr.de/ndmc).

**Table 1.** Summary of the determined wave parameters identified in Fig. 9. Wave structure (I) is separated into (I.1), lasting until 21:54 UTC and (I.2), emerging at 21:54 UTC. $\lambda$ is the horizontal wavelength once determined in position space $\lambda_1$ (see Fig.9 (a) and (b), respectively), once in frequency domain $\lambda_2$ ((c) and (d)). $\Delta t$ is the lifetime determined in the frequency domain ((c) and (d)), $T$ is the wave period (determined from (a) and (b)) and $v$ the horizontal phase velocity referring to $\lambda_1$ and $\lambda_2$.

|  | $\lambda_1$ (km) | $\lambda_2$ (km) | $\Delta t$ (min) | $T$ (s) | $v_1\left(\frac{m}{s}\right)$ | $v_2\left(\frac{m}{s}\right)$ |
|---|---|---|---|---|---|---|
| wave (I.1) | 7.0±2 | 6.8 (6.4-7.2) | 18 | 1400±100 | 5.0±1.8 | 4.9±0.6 |
| wave (I.2) | - | 9.0 (8.4-9.9) | 2 | - | - | - |
| wave (II) | 20.2±2 | 24 (20-36) | 35 | 585±100 | 34.5±9.3 | 39.3±21 |

[Figure]

**Figure 1.** Sketch of the instrument and measurement setup. See text for further details.

[Figure]

**Figure 2.** Measurement setup with a zenith angle of $45°$ viewed from the side and from aerial perspective. The axis show the definition of the coordinate system used to reference the pixels to geographical coordinates. The instrument is located at the center of the coordinate system, $z = 87 \, \text{km}$ is the altitude of the OH airglow layer, $\gamma$ is the zenith angle of the instrument and $\theta$ is the variable in the range of $\gamma \pm \text{FOV}_{\text{vertical}}$, FOV being the aperature angle of the instrument of $10°$ (vertical) and $12°$ (horizontal).

[Figure]

**Figure 3.** Mean spatial resolution of the acquired images as a function of zenith angle (black). The grey dashed line shows the respective area of the trapezium shaped FOV with the axis on the right side of the graph. The standard angle of $45°$ is marked by thin dashed lines for both curves showing a mean spatial resolution of $200 \, \text{m}$ and a corresponding area of $3400 \, \text{km}^2$.

[Figure]

**Figure 4.** (a) shows the flat-field corrected image which is distorted due to the geometry of the measurement setup with a zenith angle of $45°$. In (b) this distortion is corrected by Eq. (1). Additionally, the image is mirrored on the middle axis to change the ground-based observer's view to a satellite perspective. (c) shows the position of the FOV within the Alpine region (source: opentopomap.org, (Okt, 2015)).

[Figure]

**Figure 5.** Keogram for one row and one column of images for the night of 3rd to 4th October 2014. Between 17:30 UTC and 00:40 UTC stars can be seen clearly, but after 00:40 UTC dense cloud cover appears. Several wave events can easily be identified at 18:30, 20:00-21:00, 22:00 and 23:30 until 00:30 UTC. The time interval between 21:05 UTC and 22:30 UTC is further investigated in section 3 and 4.

[Figure]

**Figure 6.** Intensity time series for the night of 3rd to 4th October 2014. The black line shows the mean value over all pixels of FAIM 1 within the FOV of GRIPS 13 and the grey line shows the intensity measured with GRIPS 13 integrated between 1500 nm and 1600 nm. Both data sets are averaged to one minute mean values to avoid effects of different sampling rates and normalised to their individual maximum. Top: The entire time interval from the start of the measurement until 00:00 UTC, when clouds emerged. A correlation of only 0.15 is determined. Bottom: Same as before, but the first 25 minutes of twilight are avoided. The correlation now increases to 0.87.

[Figure]

**Figure 7.** Measurements in parallel direction of FAIM 1 with spectral range from 950 nm to 1650 nm, GRIPS 13 adjusted for measuring 1.27 $\mu$m $O_2$(0-0)-transition and GRIPS 16 for the OH(3-1)-Q-branch-transition at 1.51 $\mu$m for the first 3 hours of the night. The case study shows an exponential decay of $O_2$ and an increase of OH intensity. The FAIM time series (averaged over the FOV of GRIPS 13) shows a mixture of both behaviours.

[Figure]

**Figure 8.** Series of consecutive images within the chosen time interval from 21:05 UTC to 22:30 UTC. The black dashed line indicates the transverse section through the images used to analyse the smaller scale wave (I). The grey dotted line shows a transverse section approximately in direction of the wave vector for the investigation of a faint larger wave structure (II) almost perpendicular to the first one. The whole observed area is about 55 km (central width of the FOV) times 60 km (height of the FOV).

[Figure]

**Figure 9.** (a) and (b) show the temporal evolution of the transverse sections of wave (I) and wave (II). The lines indicate the position of the wavefronts. The ascissa corresponds to the position within the transverse sections shown in Fig. 8 (axis origin corresponds to the upper part of the transverse sections). Panels (c) and (d) show the Fourier transforms of each line of (a) and (b), respectively. The white areas are not significant on a 95%-level.

[Figure]

**Figure 10.** Upper panel: One minute running averages of the time series of GRIPS 13 measuring in parallel direction with FAIM 1. A smooth spline is drawn as thick line to guide the eye and show a long-period wave structure which dissolves at the end of the shown interval. Bottom panel: Fourier transform of the temperature time series. Periods of 2500, 1000, 510 and 390 s are visible in the range of higher periods. The dashed line gives the 0.95 significance. See text for further details.